# Effects of statins on $T_H1$ modulating cytokines in human subjects

Thomas R. Cimato and Beth A. Palka

Department of Medicine, State University of New York at Buffalo School of Medicine and Biomedical Sciences, Clinical and Translational Research Center, Buffalo, NY, USA

## ABSTRACT

**Background.** Activation of the innate immune system by cholesterol accelerates atherosclerosis. High levels or modified forms of cholesterol stimulate release of the inflammatory cytokines IL-12 and IL-18 that synergistically stimulate T lymphocytes to produce the atherogenic cytokine interferon-$\gamma$. While activation of the innate immune system by cholesterol is well-described in animal models and human subjects with high cholesterol levels or known atherosclerotic disease, the interaction of cholesterol and lipoproteins with the innate immune system in human subjects without known atherosclerosis is less well-described. The goal of our study was to assess the $T_H1$ modulating cytokines IL-12 p40 and IL-18, and their counter regulatory cytokines IL-18 binding protein and IL-27, to determine if their levels are linked to cholesterol levels or other factors.

**Methods.** We performed a blinded, randomized hypothesis-generating study in human subjects without known atherosclerotic disease. We measured serum lipids, lipoprotein levels, and collected plasma samples at baseline. Subjects were randomized to two weeks of therapy with atorvastatin, pravastatin, or rosuvastatin. Lipids and cytokine levels were measured after two weeks of statin treatment. Subjects were given a four-week statin-free period. At the end of the four-week statin-free period, venous blood was sampled again to determine if serum lipids returned to within 5% of their pre-statin levels. When lipid levels returned to baseline, subjects were again treated with the next statin in the randomization scheme. IL-12, IL-18, IL-18 binding protein, and IL-27 were measured at baseline and after each statin treatment to determine effects of statin treatment on their blood levels, and identify correlations with lipids and lipoproteins.

**Results.** Therapy with statins revealed no significant change in the levels of IL-12, IL-18, IL-18 binding protein or IL-27 levels. We found that IL-18 levels positively correlate with total cholesterol levels ($r^2 = 0.15$, $p < 0.03$), but not HDL or LDL cholesterol. In contrast, IL-12 p40 levels inversely correlated with total cholesterol ($r^2 = -0.17$, $p < 0.008$), HDL cholesterol ($r^2 = -0.22$, $p < 0.002$), and apolipoprotein A1 ($r^2 = -0.21$, $p < 0.002$). Similarly, IL-18 binding protein levels inversely correlated with apolipoprotein A1 levels ($r^2 = -0.13$, $p < 0.02$).

**Conclusions.** Our findings suggest that total cholesterol levels positively regulate IL-18, while HDL cholesterol and apolipoprotein A1 may reduce IL-12 p40 and IL-18 binding protein levels. Additional studies in a larger patient population are needed to confirm these findings, and verify mechanistically whether HDL cholesterol can directly suppress IL-12 p40 and IL-18 binding protein levels in human subjects.

Corresponding author
Thomas R. Cimato,
tcimato@buffalo.edu

## INTRODUCTION

Cholesterol activates innate immune responses via two interrelated mechanisms (Fig. 1). Modified LDL cholesterol particles are absorbed into cells through the scavenger receptors (CD36, SR-A1, SR-A2, MARCO, SR-B1, LOX-1, and PSOX) (*Hansson & Hermansson, 2011*). Accumulation of intracellular cholesterol can lead to crystal formation. Cholesterol crystals, in combination with other inflammatory stimuli such as lipopolysaccharide, activate NLRP-3 inflammasomes. This results in caspase-1 dependent release of IL-1$\beta$ and IL-18, which both promote atherosclerosis (*Duewell et al., 2010*). In the second mechanism, oxidized LDL binds to the family of Toll-like receptors (TLRs). TLR signaling via the adaptor protein MyD88 activates a signaling cascade that leads to expression of genes encoding pro inflammatory cytokines IL-6, IL-12 and TNF alpha (*Hansson & Hermansson, 2011*). These two pathways activated by oxidized or modified LDL cholesterol produce the atherogenic cytokines IL-12 and IL-18.

T-helper-1 ($T_H1$) cells drive athersosclerosis. Differentiation of naive T cells to the $T_H1$ phenotype is synergistically induced by IL-12 and IL-18 (Fig. 2). IL-12 and IL-18 stimulate $T_H1$ cells to produce interferon-$\gamma$, which promotes atherosclerosis by activation of macrophages, inducing production of pro-inflammatory mediators, and increasing smooth muscle cell proliferation, collagen production, and matrix metalloproteinase (MMP) activity (*Hansson & Hermansson, 2011*). In humans, IL-12 and IL-18 are associated with cardiovascular disease events and mortality (*Hansson & Hermansson, 2011*; *Rabkin, 2009*). The IL-18 binding protein (IL-18BP) is a secreted protein that binds IL-18 but lacks features of a receptor, and is normally present at ten-fold higher levels than IL-18 in human subjects (*Dinarello et al., 2013*). In contrast to the atherogeneic effects of IL-18, IL-18BP is an endogenous modulator of IL-18 that prevents fatty streak development and slows progression of advanced atherosclerotic plaques in murine models of atherosclerosis (*Mallat et al., 2001*). Similarly, IL-27 is also a member of the IL-6/IL-12 superfamily of cytokines. IL-27 induces the IL-12 receptor expression, and elicits secretion of interferon-$\gamma$ (*Hunter & Kastelein, 2012*). As a result, IL-27 influences differentiation of naive T cells to $T_H1$ cells while inhibiting formation of $T_H17$ and $T_H2$ cells. The lack of IL-27 receptor (*Koltsova et al., 2012*) and IL-27 p28 subunit (*Hirase et al., 2013*) expression in animal models accelerates atherosclerosis, indicating an anti-atherogenic role for IL-27 by reducing recruitment of myeloid cells into atherosclerotic vessels.

Given the clear role of cholesterol in naive T cell polarization to $T_H1$ cells via IL-12 and IL-18 production, and the anti-atherosclerotic functions of IL-18BP and IL-27, we conducted an hypothesis-generating study to determine if IL-12, IL-18, IL-18BP, and IL-27 levels varied with cholesterol levels in human subjects without known atherosclerotic disease. In this study we determined baseline IL-12, IL-18, IL-18BP, and IL-27 levels and

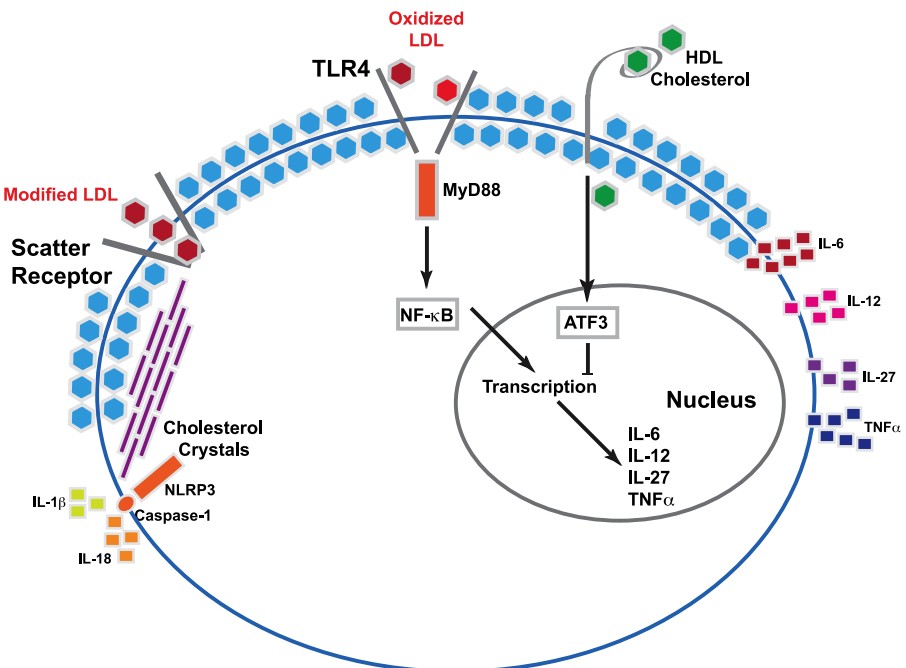

**Figure 1 Molecular mechanism underlying cholesterol stimulated cytokine release.** Modified LDL cholesterol is absorbed into cells via scatter receptors. At high intracellular concentrations of cholesterol, crystals form. In combination with inflammatory stimuli such as lipopolysaccharide, cholesterol crystals activate NLRP3 inflammasome-caspase-1 activity, leading to the release of mature IL-1b and IL-18. The Toll-like family receptors recognize oxidized LDL. Binding of oxidized LDL cholesterol activates inflammatory signaling via MyD88 and NF$\kappa$B. This results in increased transcription of the inflammatory cytokines IL-6, IL-12, IL-27 and TNF$\alpha$. HDL cholesterol alters the activity of ATF3 to suppress TLR activated transcription of inflammatory cytokines.

after modulation of cholesterol levels using statins with different potencies to determine if these pro- and anti-atherosclerotic cytokines are linked to cholesterol levels, lipoproteins, or other factors.

# MATERIALS AND METHODS

## Patient consent for participation

Informed consent to undergo the study protocol was obtained in writing from each study subject according to the principles expressed in the Declaration of Helsinki and approved by the University at Buffalo Institutional Review Board for Health Sciences Research (Approval Number: MED5980509B).

## Characteristics of study subjects

The study population consisted of 12 adult subjects (7 males and 5 females) with no medical problems. Study subjects were screened to exclude subjects with chronic health disorders that may impact the study results including hypercholesterolemia (total cholesterol >300 mg/dL or already on statin treatment) with additional cardiovascular disease risk factors, cancer, diabetes, chronic liver or kidney disease, heart failure or diseases of chronic inflammation. The baseline characteristics of the study population

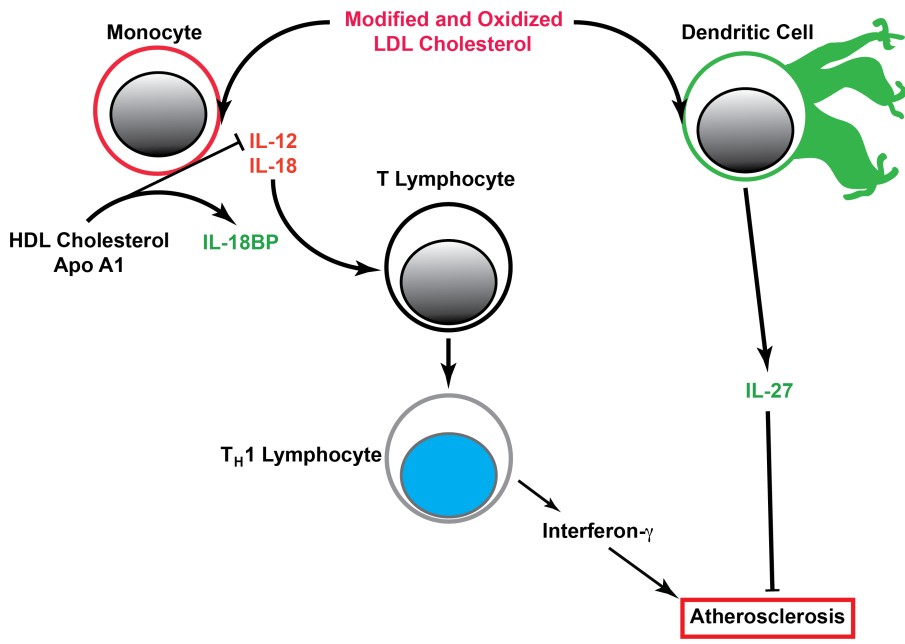

**Figure 2 Cellular mechanism of cholesterol induced T$_H$1 lymphocyte differentiation and atherosclerosis.** Dendritic cells and monocytes recognize modified and oxidized LDL cholesterol. In monocytes, this results in release of IL-12 and IL-18 that act on T lymphocytes to promote polarization to T$_H$1 lymphocytes, and promote release of interferon-$\gamma$. HDL cholesterol and recombinant HDL particles containing apolipoprotein A1 (ApoA1) suppress transcription of IL-12. IL-18 binding protein (IL-18BP) absorbs IL-18, modulating T$_H$1 activation and interferon-$\gamma$ release. Interferon-$\gamma$ augments atherosclerosis. Oxidized LDL promotes release of IL-27, which attenuates atherosclerosis.

were reported previously (*Cimato et al., 2013*; *Cimato & Palka, 2014*) and are shown in Table 1. The age of the cohort was 43.4 ± 12.5 years. The average body mass index was 24.9 ± 7.2. The average Framingham Risk Score was 1.7 ± 0.5 indicating low risk for atherosclerotic disease events in the cohort. Prior to statin treatment the mean total cholesterol level was 210.5 ± 27.6 mg/dL, LDL cholesterol 136.2 ± 22.9 mg/dL, HDL cholesterol 53.5 ± 12.9 mg/dL. The cohort had a low index of inflammation, as the C-reactive protein level was 1.1 ± 1.3 mg/L. Two of the study subjects had treated hypertension.

## Study protocol

Study subjects were randomized to drug regimen groups using a block randomization design. The investigators were blinded to which treatment subjects were receiving. Study subjects underwent a baseline blood draw in which a complete blood cell count, lipid panel (total, HDL, and LDL cholesterol, and triglycerides) and C-reactive protein were determined by the Kaleida Health pathology laboratory. The plasma fraction was also retained from each blood draw and frozen at −80 °C for chemokine and cytokine assays. Subjects were then treated for two weeks with one of three different HMG-CoA reductase inhibitors (atorvastatin 80 mg daily, pravastatin 80 mg daily, or rosuvastatin 10 mg daily). At the end of the two-week statin treatment, venous blood was sampled again to obtain the

**Table 1 Characteristics of the patient cohort.** Age, BMI, Framingham Risk Score, serum lipid levels and C-reactive protein levels are provided at baseline and after treatment for two weeks with Atorvastatin, Pravastatin, and Rosuvstatin.

| | Baseline | Atorvastatin | Pravastatin | Rosuvastatin |
|---|---|---|---|---|
| **Age** | 43.4 ± 12.5 | | | |
| **BMI** | 24.9 ± 7.2 | | | |
| **Framingham Risk Score** | 1.7 ± 0.5 | | | |
| **Total cholesterol (mg/dL)** | 210.5 ± 27.6 | 138.5 ± 28.9[*] | 160.6 ± 28.9[*] | 154.2 ± 21.0[*] |
| **LDL cholesterol (mg/dL)** | 136.2 ± 22.9 | 68.2 ± 12.1[*] | 88.7 ± 24.0[**] | 83.3 ± 12.7[*] |
| **HDL cholesterol (mg/dL)** | 53.5 ± 12.9 | 54.1 ± 18.6 | 54.5 ± 13.1 | 55.3 ± 15.3 |
| **C-reactive protein (mg/L)** | 1.1 ± 1.3 | 1.05 ± 1.1 | 0.95 ± 0.8 | 1.3 ± 1.5 |

**Notes.**

Values are Mean ± SD.

[*] $p < 0.0001$ vs. baseline.

[**] $p < 0.0002$ vs. baseline.

lipid panel, C-reactive protein level, and blood samples for cytokine analysis. Subjects were subsequently given a four-week statin-free period. At the end of the four-week statin-free period, venous blood was sampled again to determine if serum lipids returned to within 5% of their pre-statin levels. In subjects where cholesterol levels did not recover to within 5% of the baseline lipid levels, an additional four-week statin-free period was provided before resuming statin therapy to avoid effects of overlap between drugs. Three study subjects required extension of the statin-free period for an additional four weeks for serum lipid levels to return to within 5% of their baseline lipid levels. Following this period, the next HMG-CoA reductase inhibitor in the randomization scheme was given for two weeks. The same protocol was repeated for statin drugs two and three until study completion. All twelve subjects completed treatment with the three statin medications.

## Apolipoprotein and lipoprotein (a) assays

Apolipoprotein A1, B, and Lp(a) levels were determined using the VAP test (Atherotech Diagnostics Lab, Birmingham, AL, USA) on plasma from human subjects.

## Cytokine assays

Plasma levels of the following cytokines were measured using ELISA assays: IL-12 p40 (R&D Systems, Catalog # DP400), IL-18 (eBioscience, Catalog # BMS267/2), IL-18BPa (R&D Systems, Catalog # DBP180), and IL-27 (Abcam, Catalog # ab83695). The intra-assay CV% for each assay is the following: IL-12 p40-6.4; IL-18-6.5; IL-18BPa-3.9; IL-27-4.6. The inter-assay CV% for each assay is the following: IL-12 p40: 6, IL-18: 8.1; IL-18BPa: 8.9; IL-27: 7.8. Plasma levels of IL-12 p40 and IL-18 were performed in triplicate, IL-18BP and IL-27 performed in singulate measurements per subject, per treatment.

## Statistical analysis

Regression analyses were tested for significance using Pearson's correlation. Residual analysis confirmed normal distribution of residuals for each cytokine across subjects, and confirmed the data sets were appropriate for linear regression analysis. Significant

differences in IL-12 p40, IL-18, IL-18BP, and IL-27 plasma levels between statin therapies were assessed using a non-parametric Friedman's two-way ANOVA by ranks with a Bonferroni correction for multiple comparisons. In cases where cytokine levels were undetectable for any of the experimental conditions, all data for that subject were not included in the ANOVA analysis. The data used for the ANOVA analysis of effects of statins are contained in the Data S1. Data used for linear regression analysis is included in the Data S2. Numerical data stated in the manuscript text represent mean $\pm$ standard error of the mean. Statistical analysis was performed using SPSS software. Box plots shown in the manuscript were generated using BoxPlotR (*Spitzer et al., 2014*).

## RESULTS

### Statin therapy does not significantly alter IL-12, IL-18, IL-18BP or IL-27 levels in human subjects

To assess whether the atherogenic cytokines IL-12 and IL-18 vary with cholesterol levels in humans, we measured IL-12 and IL-18 plasma levels in human subjects without known coronary disease and used HMG-CoA reductase inhibitors (statins) of varying potencies to lower cholesterol levels from baseline. The baseline lipid levels of the cohort are shown in Table 1 and published previously (*Cimato et al., 2013*; *Cimato & Palka, 2014*). Therapy with statins significantly reduced total and LDL cholesterol levels. We found no significant differences between IL-12 p40 levels (Fig. 3A) or IL-18 levels (Fig. 3B) and statin treatments, $p > 0.05$. Next we determined the effects of statin therapy on anti-atherogenic factors IL-18BP and IL-27. We again found no significant differences between IL-18BP levels (Fig. 4A) or IL-27 levels (Fig. 4B) and statin treatments, $p > 0.05$. Given the small number of subjects studied and the lack of detectable cytokine in some subjects, our data on the effect of statins on these cytokines are inconclusive and require a larger subject cohort to answer the question properly.

### Total cholesterol positively correlates with IL-18 and negatively correlates with IL-12 p40 levels

We next analyzed the data set to identify relationships between serum lipids and cytokines. Linear regression analyses (Table 2) identified a significant, negative correlation between total cholesterol levels and IL-12 ($r^2 = -0.17, p < 0.008$, Fig. 5A).

We also noted a significant, positive correlation between total cholesterol levels and IL-18 ($r^2 = 0.15, p < 0.03$, Fig. 5B). We found no significant correlation between IL-18 levels and LDL or HDL cholesterol. No correlations were identified between total cholesterol levels and IL-18BP or IL-27. The findings suggest total cholesterol, or components contained in the total cholesterol fraction, augment IL-18 levels in human subjects.

### HDL cholesterol and apolipoprotein A1 negatively correlate with IL-12 levels

Ligands bound by Toll-like receptors (TLRs) such as oxidized LDL cholesterol induce the pro-inflammatory and atherogenic cytokines IL-12 (*Hansson & Hermansson, 2011*), and IL-27 (*Bosmann & Ward, 2013*) (Fig. 1). A recent study showed that HDL cholesterol

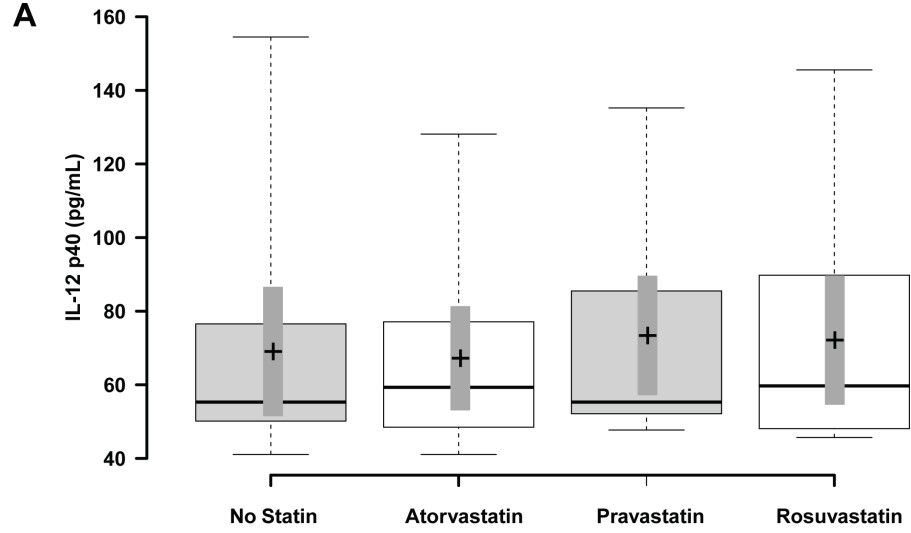

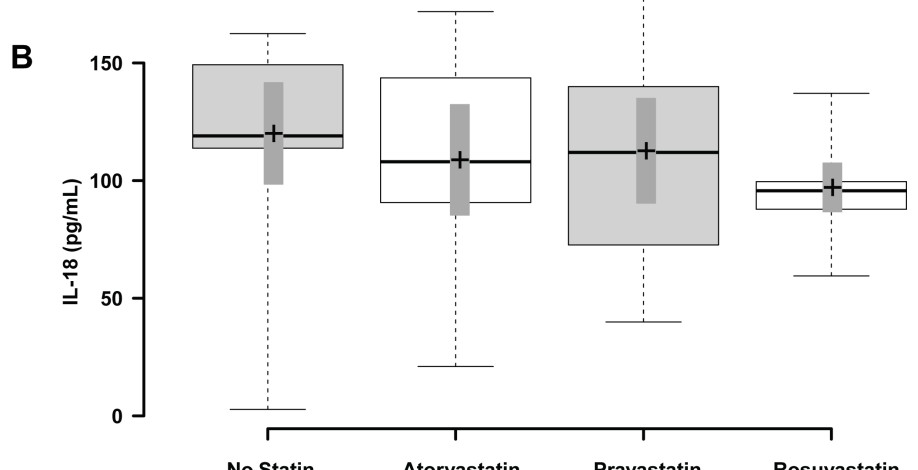

**Figure 3 The effect of statin treatment on IL-12 p40 and IL-18 levels in human subjects.** IL-12 p40 subunit (A) and IL-18 (B) levels were measured by ELISA at baseline and after two weeks of treatment with atorvastatin, pravastatin, or rosuvastatin. No significant differences were noted between statin treatments on IL-12 p40 or IL-18. Data are presented as box plots. Center lines show the medians; box limits indicate the 25th and 75th percentiles as determined by R software; whiskers extend to minimum and maximum values; crosses represent sample means; bars indicate 90% confidence intervals of the means. $n = 11$ subjects for IL-12 p40, $n = 8$ subjects for IL-18.

suppresses the effects of TLR pathway signaling on pro-inflammatory cytokine production via ATF3 (*De Nardo et al., 2014*). The study by De Nardo et al. indicates HDL cholesterol reduces the levels of IL-12 in the setting of co-stimulation of the TLR receptor. Our cohort had relatively low levels of C-reactive protein, suggesting minimal inflammation and was screened to exclude subjects with diseases associated with inflammation. Given this background, we tested whether the levels of TLR-induced cytokines correlated with HDL

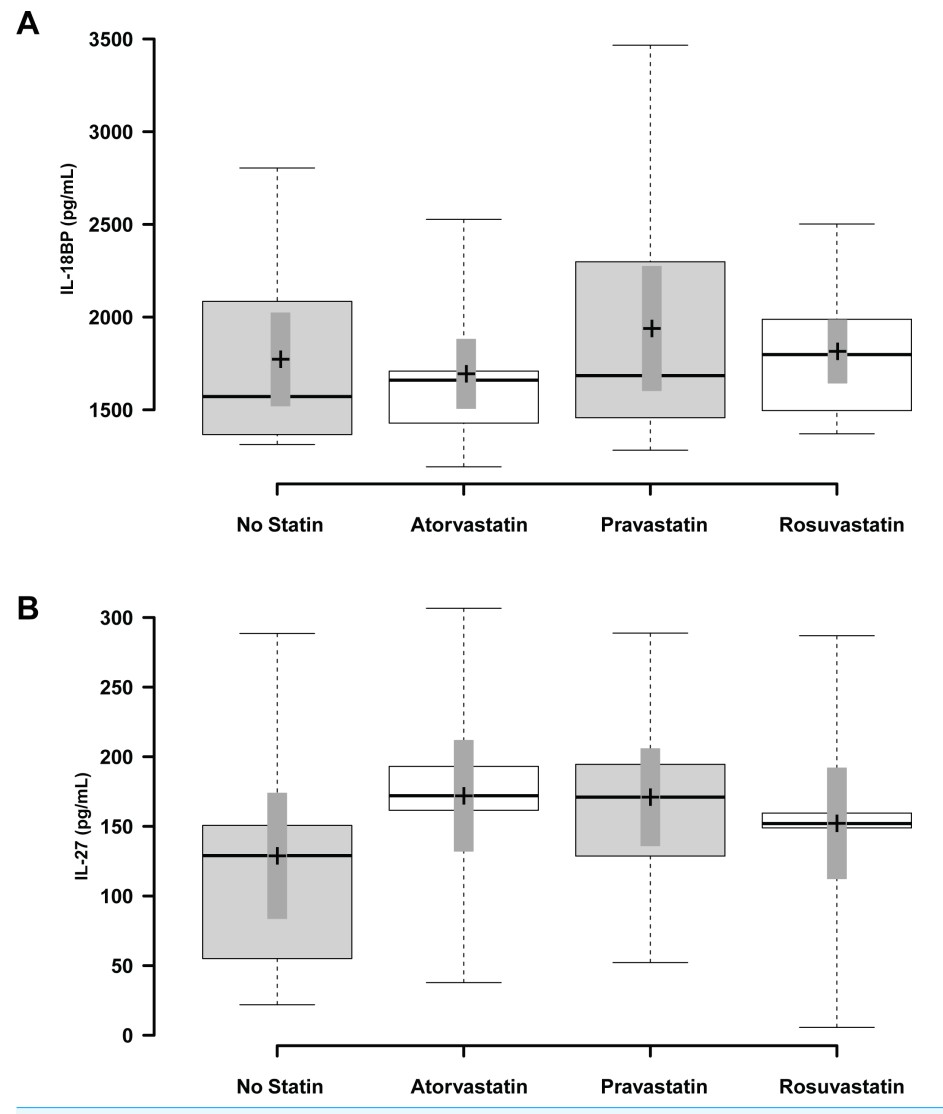

**Figure 4  The effect of statin treatment on IL-18BP and IL-27 levels in human subjects.** IL-18BP (A) and IL-27 (B) levels were measured by ELISA at baseline and after two weeks of treatment with atorvastatin, pravastatin, or rosuvastatin. No significant differences were noted between statin treatments on IL-18BP or IL-27. Data are presented as box plots. Center lines show the medians; box limits indicate the 25th and 75th percentiles as determined by R software; whiskers extend to minimum and maximum values; crosses represent sample means; bars indicate 90% confidence intervals of the means. $n = 11$ subjects for IL-18BP, $n = 5$ subjects for IL-27.

cholesterol or apolipoprotein A1. We found significant negative correlation between HDL cholesterol and IL-12 p40 levels ($r^2 = -0.22$, $p < 0.002$, Fig. 6A), and apolipoprotein A1 and IL-12 p40 levels ($r^2 = -0.21$, $p < 0.002$). Additionally, we found significant correlation between apolipoprotein A1 and IL-18BP ($r^2 = -0.13$, $p < 0.02$, Fig. 6B), but not with HDL cholesterol. No relationship was found between HDL cholesterol or apolipoprotein A1 and IL-18 or IL-27. The findings suggest a role for HDL cholesterol mediated suppression of TLR-stimulated inflammatory cytokine production, including IL-12 p40 and IL-18BP.

**Table 2 Pearson's correlation analysis of serum lipids, lipoproteins, and cytokines.** Linear regression analysis measured correlations between lipid and lipoprotein levels, and IL-12 p40, IL-18, IL-18BP, and IL-27. The $p$-values and $r^2$ values are shown. Statistically significant correlations are indicated in bold type.

| | $p$-value | | | | $r^2$ value | | | |
|---|---|---|---|---|---|---|---|---|
| | **IL-18** | **IL-18BP** | **IL-12 p40** | **IL-27** | **IL-18** | **IL-18BP** | **IL-12 p40** | **IL-27** |
| **Total cholesterol** | **0.022** | 0.31 | **0.007** | 0.42 | **0.15** | 0.02 | **−0.17** | 0.02 |
| **LDL cholesterol** | 0.089 | 0.88 | 0.44 | 0.99 | 0.09 | 0.0005 | 0.015 | 0 |
| **HDL cholesterol** | 0.16 | 0.1 | **0.0009** | 0.25 | 0.06 | −0.06 | **−0.24** | 0.05 |
| **Apo A1** | 0.17 | **0.02** | **0.001** | 0.16 | 0.06 | −0.12 | **−0.24** | 0.07 |
| **Lp(a)** | 0.93 | 0.87 | **0.02** | **0.003** | 0.0002 | 0.0006 | **−0.12** | **−0.29** |

**Lipoprotein A negatively correlates with IL-12 and IL-27 levels**

Lipoprotein A (Lp(a)) is an LDL-like particle that is covalently bound to apolipoprotein B in an LDL-like particle. The function of Lp(a) is largely unknown but has a high degree of homology with plasminogen, suggesting a role in coagulation (*McLean et al., 1987*). High levels of Lp(a) are associated with higher rates of cardiovascular disease events (*Nordestgaard et al., 2010*), but very little is known about the role of Lp(a) in stimulation of innate immune responses. Lp(a) negatively correlated with both IL-12 p40 ($r^2 = -0.12$, $p < 0.05$, Fig. 7A), and IL-27 ($r^2 = -0.29$, $p < 0.003$, Fig. 7B). No significant correlation was found between Lp(a) and IL-18 or IL-18BP. The correlation between Lp(a) and IL-12 p40 is similar to the findings with total cholesterol levels, and appears to have an effect on IL-12 p40 levels similar to HDL cholesterol. Lp(a) was the only blood lipid component to correlate significantly with IL-27 levels, suggesting Lp(a) has an effect in decreasing IL-27 levels. The relationships between Lp(a), IL-12 p40, and IL-27 levels have not been described previously to our knowledge.

## DISCUSSION

Our hypothesis-generating study contributes preliminary evidence suggesting interactions between serum lipids and cytokines tied to atherogenesis in normal human subjects without known atherosclerosis. We conclude the following: (1) Our study is underpowered to determine if statin treatment effects IL-12 p40, IL-18, IL-18BP or IL-27 levels; (2) IL-18 levels positively correlate with total cholesterol levels; (3) IL-12 p40 levels inversely correlate with HDL cholesterol, apolipoprotein A1, and Lp(a); (4) IL-18BP levels inversely correlate with apolipoprotein A1 levels; (5) IL-27 levels inversely correlate with Lp(a).

We found no significant effect of statin treatment on the levels of IL-12 p40, IL-18, IL-18BP or IL-27 (Figs. 3 and 4) in normal human subjects; however, our data set is underpowered to test this question adequately. Prior studies have shown that statins can actually increase the levels of both the NLRP3-inflammosome linked cytokines IL-1$\beta$ (*Kuijk et al., 2008*; *Liao et al., 2013*) and IL-18 (*Coward et al., 2006*) in the presence of co-stimulants, including lipopolysaccharide or PHA. Similarly, simvastatin was shown

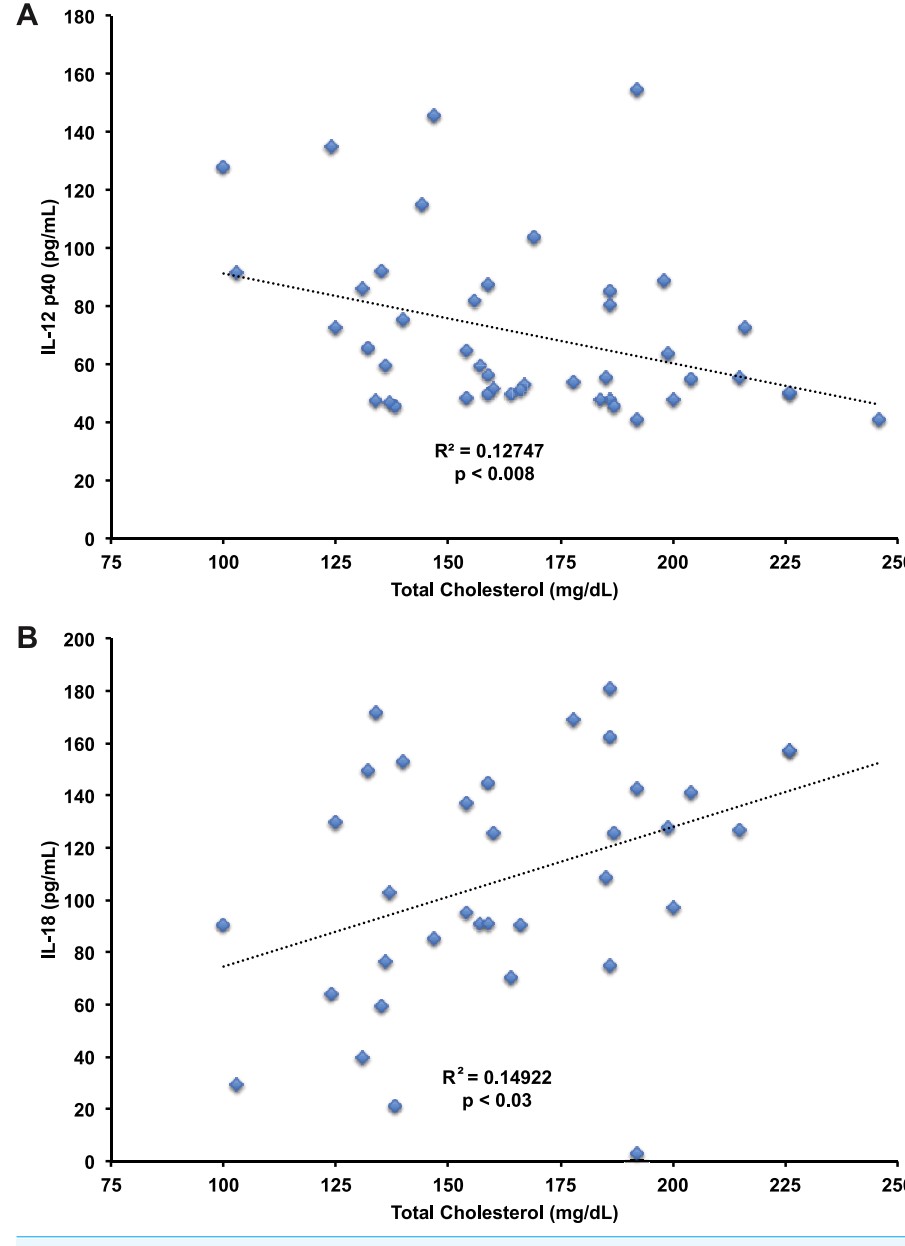

**Figure 5 Correlation of IL-12 p40 and IL-18 with total cholesterol levels.** Pearson's correlation of IL-12 p40 ($r^2 = -0.17$, $p < 0.008$, A) and IL-18 ($r^2 = 0.15$, $p < 0.03$, B) levels in plasma with total cholesterol levels from human subjects. IL-12 p40 and IL-18 data in in $n = 11$ subjects at baseline and after two weeks treatment with atorvastatin, pravastatin, or rosuvastatin. IL-12 p40 was detected in all study samples, IL-18 detected in 39 of 44 study samples.

to increase IL-12 p40 in conjunction with lipopolysaccharide in monocytes (*Matsumoto et al., 2004*). We did not observe any effect of statins to induce cytokines in our normal human subject cohort likely due to a low level of inflammation. It would be of interest to determine if the effect of statins on IL-12 and IL-18 levels are a class effect of the drugs, and if statin treatment of subjects with the inflammatory stimulus of coronary artery

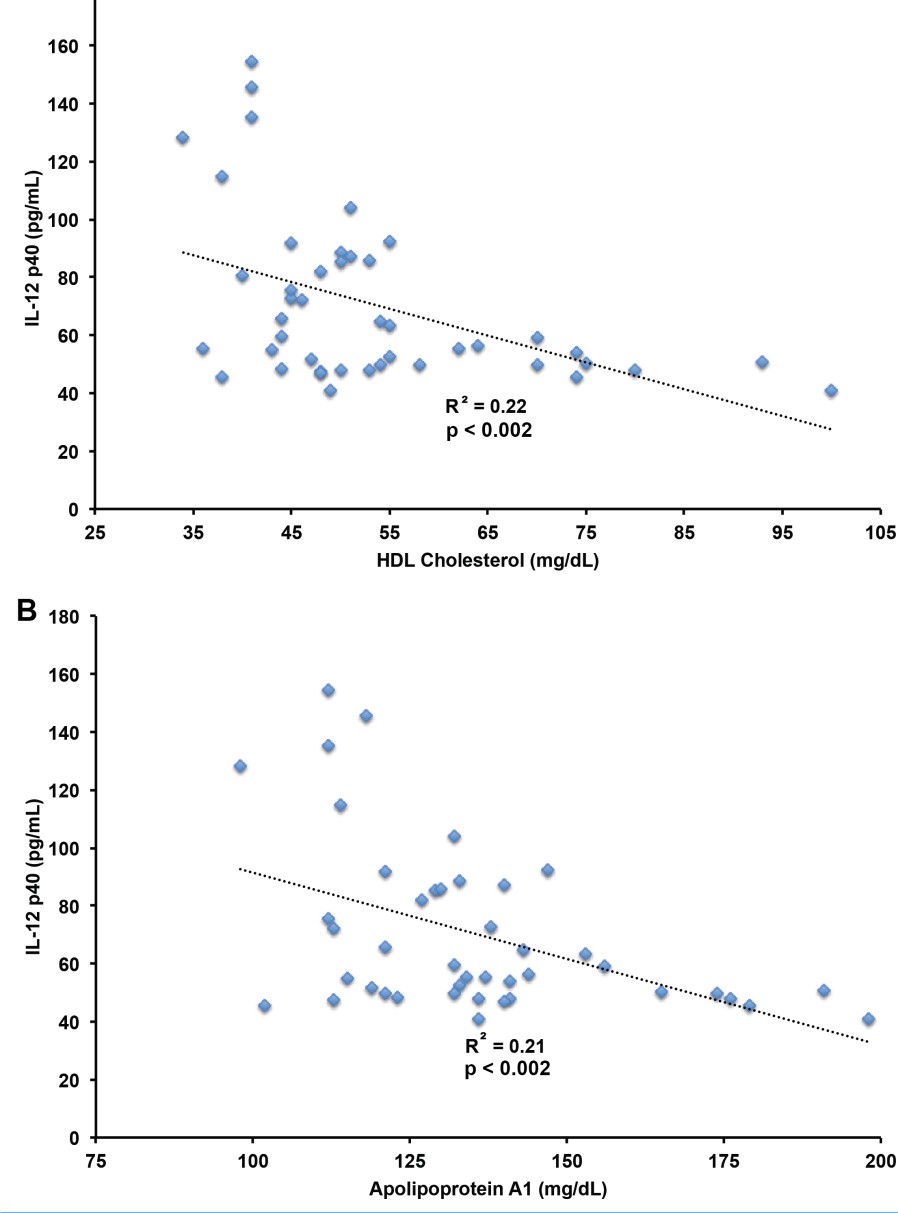

**Figure 6 Correlation of IL-12 p40 with HDL cholesterol and apolipoprotein A1 levels.** Pearson's correlation of IL-12 p40 with HDL cholesterol levels ($r^2 = -0.22$, $p < 0.002$, A), and apolipoprotein A1 levels ($r^2 = -0.21$, $p < 0.002$, B) in plasma from human subjects. IL-12 p40, HDL cholesterol, and apolipoprotein A1 data in in $n = 11$ subjects at baseline and after two weeks treatment with atorvastatin, pravastatin, or rosuvastatin. HDL cholesterol and IL-12 p40 was detected in all study samples, apolipoprotein A1 detected in 43 of 44 study samples.

disease results in increased IL-12 and IL-18 release. Studies showing increased cytokine release with statins were performed *in vitro*, and may be due to effects of the drugs at supra-physiologic concentrations in the micromolar range (*Matsumoto et al., 2004*). The $IC_{50}$ for simvastatin in cultured cells ranges up to 10 nM, suggesting that the observed activation of simvastatin on IL-18 release may be occurring at higher concentrations of

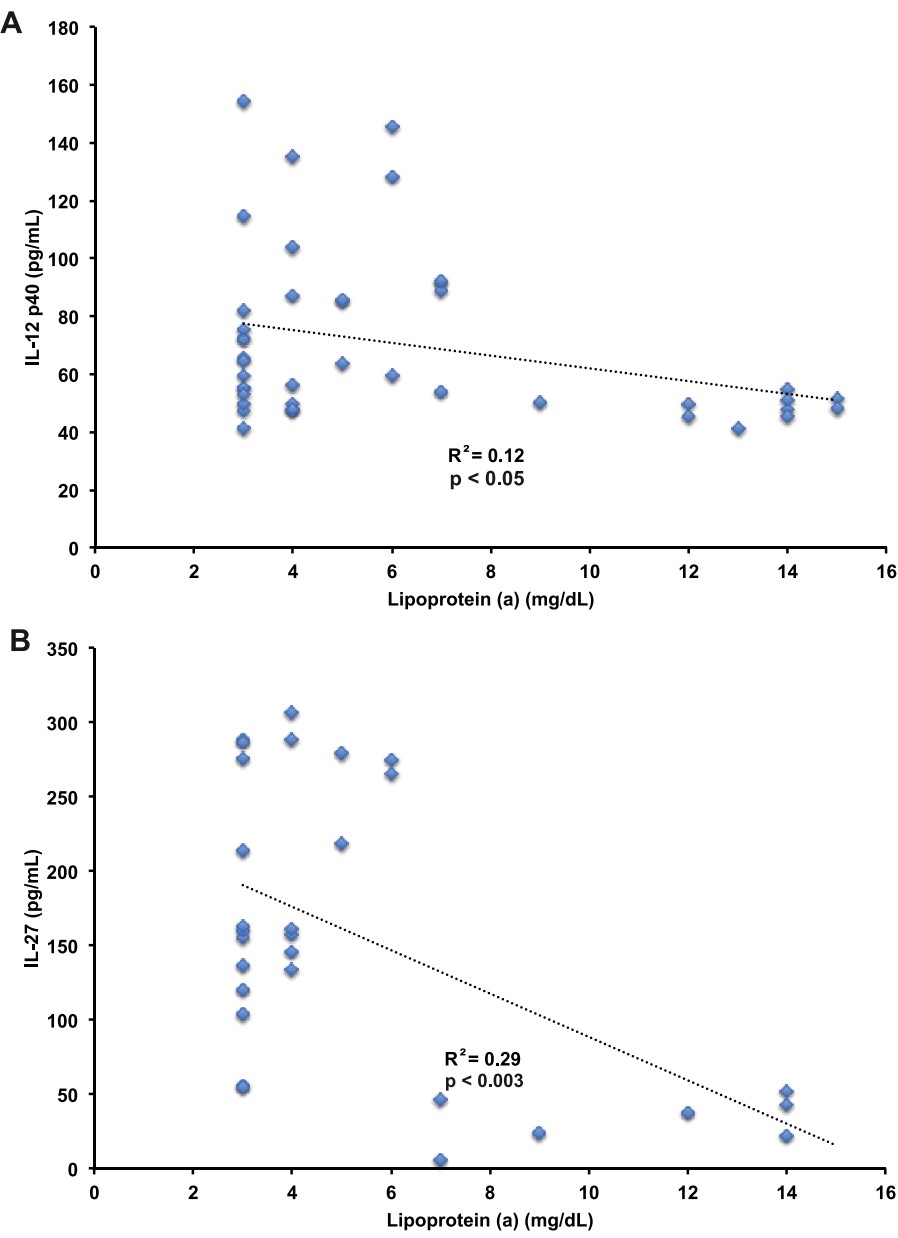

**Figure 7 Correlation of IL-12 p40 and IL-27 with lipoprotein (a).** Pearson's correlation of IL-12 p40 with lipoprotein (a) levels ($r^2 = -0.12$, $p < 0.05$, A), and IL-27 with lipoprotein (a) levels ($r^2 = 0.29$, $p < 0.003$, B) in plasma from human subjects. IL-12 p40, IL-27, and lipoprotein (a) data in $n = 11$ subjects at baseline and after two weeks treatment with atorvastatin, pravastatin, or rosuvastatin. IL-12 p40 was detected in all study samples. Lipoprotein (a) detected in 43 of 44 study samples. IL-27 detected in 34 of 44 study samples.

simvastatin than likely to be encountered *in vivo* (*Shitara & Sugiayama, 2006*). We focused our analysis to identify relationships between cholesterol levels and cytokines using statins to modulate cholesterol levels. However, it should also be noted that the statins themselves might also play a role in modulating cytokine levels. While our patient cohort was too

small to observe significant differences in cytokine levels between statin treatments, others have found effects of different statins on cytokine production both *in vitro* and in mice (*Bessler et al., 2005*; *Djaldetti et al., 2006*).

We identified a positive correlation between total cholesterol and IL-18 levels, but no significant interaction with HDL or LDL cholesterol, apolipoprotein A1 or B. Prior studies show a variety of correlations between IL-18 and serum lipid levels. In pre-menopausal women, IL-18 levels negatively correlated with LDL cholesterol size and positively correlated with HDL cholesterol levels (*Berneis et al., 2010*). In the Atherosclerosis Risk in Communities Study (ARIC) IL-18 levels in Caucasians and African-Americans had significantly different correlations with serum lipid levels (*Negi et al., 2012*). Caucasian subjects showed positive correlation between IL-18 levels and LDL cholesterol and negative correlation between IL-18 and HDL cholesterol. In African-Americans, LDL cholesterol displayed no correlation with IL-18 levels, and HDL cholesterol negatively correlated with IL-18 levels. Our study was a much smaller cohort, focused on a younger cohort of subjects with lower body mass index, with no African-American subjects. These factors may explain our inability to detect interactions between HDL or LDL cholesterol and IL-18.

Mechanistically, it is unclear why total cholesterol but not HDL or LDL cholesterol correlated with IL-18 levels. IL-18 precursor protein is present in blood monocytes and epithelial cells of the gastrointestinal tract (*Dinarello et al., 2013*) and IL-18 release is stimulated by activation of the NLRP3 inflammasome (*Dinarello et al., 2013*). Activation of the NLRP3 inflammasome occurs via several pattern recognition receptors that detect microbial structures termed pathogenic associated molecular patterns (PAMPs), or danger-associated molecular patterns (DAMPs) in response to uric acid and cholesterol crystals, and high mobility group box 1 (HMGB1) (*Lamkanfi & Dixit, 2014*). The stimulus for IL-18 release in the total cholesterol fraction may include modified or oxidized LDL particles (Fig. 1) which were not measured in this study. Accumulation of cholesterol crystals within cells also may activate the inflammasome via DAMPs (*Lamkanfi & Dixit, 2014*). To our knowledge, the components of serum lipids most likely to form cholesterol crystals within cells have not been identified.

The strongest correlations noted from our data was the inverse correlation between HDL cholesterol, apolipoprotein A1, and IL-12 p40 levels in human subjects. A novel mechanism was identified recently where HDL cholesterol, or reconstituted HDL cholesterol consisting of apolipoprotein A1 and phospholipids, induces expression of the transcription factor ATF3, and enhances ATF3 binding to transcriptional sites. HDL cholesterol stimulated induction of ATF3 in macrophages resulted in suppression of several TLR stimulated cytokines, including IL-6, IL-12 and TNF$\alpha$ (*De Nardo et al., 2014*). IL-12 combined with IL-18 synergistically induces interferon-$\gamma$ from $T_H1$ cells, and is linked to atherogenesis (*Hansson & Hermansson, 2011*). Very few human subjects-based studies larger than ours have evaluated correlations between IL-12 levels and serum lipids. In obese Mexicans, serum IL-12 levels increased with body mass index, and correlated closely with triglyceride levels, but total cholesterol levels were not significantly associated with IL-12 (*Suarez-Alvarez et al., 2013*). In contrast, we found no relationship between IL-12, BMI, or

triglyceride levels. We may have been able to observe the effects of HDL cholesterol to lower IL-12 levels due to a lack of significant inflammation in our study cohort.

We observed negative correlation between IL-12 and Lp(a) levels, similar to the effect of HDL cholesterol. The function of Lp(a) in the pathogenesis of atherosclerosis, or its role in lipid metabolism, are poorly understood. A recent correlative study examined the relationship of Lp(a) to other lipoproteins and found very strong correlation between Lp(a), HDL cholesterol, apolipoprotein A1, and very low-density lipoprotein, but not LDL cholesterol or apolipoprotein B (*Konerman et al., 2012*). Given that IL-12 levels were negatively correlated with Lp(a) similar to our observations of HDL cholesterol, this suggests Lp(a) may have overlapping function and possibly activate ATF3 dependent suppression of cytokine production. Future experiments will determine if HDL cholesterol and Lp(a) share a common mechanism to suppress inflammatory cytokine production.

IL-18BP displays a high affinity for IL-18 ($K_d$ 399 pM) (*Kim et al., 2000*) and is present in the serum of healthy subjects at 20-fold molar excess to IL-18 to blunt the $T_H1$ response to foreign organisms and reduce autoimmune responses to routine infection (*Dinarello et al., 2013*). IL-18BP expression is highly regulated at the level of transcription by interferon-$\beta$, $\gamma$, and IL-27 in a negative feedback loop (*Dinarello et al., 2013*). In murine models of atherosclerosis, IL-18BP reduces fatty streak development, and slows progression of advanced atherosclerotic plaques in the thoracic aorta of apoE knockout mice (*Mallat et al., 2001*). In human subjects, the relationship between serum lipid levels and IL-18BP levels has not been extensively studied (*O'Brien et al., 2014*). We found that IL-18BP was negatively correlated with apolipoprotein A1 levels in human subjects ($r^2 = -0.12$, $p = 0.02$) and showed a trend towards negative correlation with HDL cholesterol. Our finding that IL-18BP is negatively correlated with apolipoprotein A1 levels suggests a similar HDL cholesterol-ATF3 dependent mechanism to suppress IL-18BP expression. Future experiments will test this possibility, as stimulation of IL-18BP expression represents a potential means to reduce atherogenic IL-18, and its downstream product interferon-$\gamma$.

We also found a significant, negative correlation between Lp(a) and IL-27 levels. IL-27 is a member of the IL-12 family of cytokines that, in contrast to IL-12 and IL-18, limits the intensity and duration of T cell responses (*Hunter & Kastelein, 2012*), and limits atherosclerosis in animal models of atherosclerosis (*Hirase et al., 2013*; *Koltsova et al., 2012*). Antigen presenting cells produce IL-27 in response to stimulation of TLRs. TLR activation via MyD88 augments IL-27 transcription (*Bosmann & Ward, 2013*) (Fig. 1). We did not observe significant correlation between IL-27 and other serum lipid components. The mechanism of the effect of Lp(a) on IL-27 has not been revealed to date. A prior study of human subjects with a spectrum of coronary artery disease showed IL-27 levels strongly, positively correlate with oxidized LDL cholesterol levels and the Gensini score (an index coronary atherosclerosis severity) (*Jin et al., 2012*). Mechanistically, oxidized LDL stimulated IL-27 production from dendritic cells in a time and concentration dependent manner. In the context of the findings noted in animal models of atherosclerosis, oxidized

LDL induced IL-27 expression likely represents a counter-regulatory mechanism to suppress inflammation in atherosclerosis. In the context of the anti-inflammatory function of IL-27, our correlative finding that Lp(a) negatively correlates with IL-27 levels suggests Lp(a) may suppress this anti-atherosclerotic cytokine, and merits further mechanistic evaluation in future experiments.

The findings of our study are limited by the small sample size of the cohort and should be considered hypothesis-generating and not definitive in nature. Our study focused on the effects of reduction of serum lipid levels by statins in healthy adult subjects. As a result, the correlations we identify with lowering of serum lipids may not be applicable in subjects with hypercholesterolemia, or other illnesses associated with atherosclerosis.

## CONCLUSIONS

Our hypothesis-generating study provides preliminary evidence for the interactions between serum lipids and $T_H1$ modulating cytokines in normal human subjects without known atherosclerosis. We draw the following conclusions: (1) IL-18 levels positively correlate with total cholesterol levels; (2) IL-12 p40 levels inversely correlate with HDL cholesterol, Apolipoprotein A1, and Lp(a); (3) IL-18BP levels inversely correlate with Apolipoprotein A1 levels; (4) IL-27 levels inversely correlate with Lp(a). While our study is limited by the small cohort size, the relationships between serum lipids components and $T_H1$ modulating cytokines revealed unique findings for the effect of HDL cholesterol, apolipoprotein A1 and Lp(a) on these cytokines. When viewed in the context of the recent discovery that HDL cholesterol activates suppression of inflammatory cytokine expression via the transcription factor ATF3 (*De Nardo et al., 2014*), our findings suggest a similar mechanism is active in human subjects. Inverse correlations observed between Lp(a), IL-12, and IL-27 suggest Lp(a) may be similar to HDL cholesterol in regulation of cytokine levels. Our future studies will verify the findings in larger and more diverse cohorts, and explore the suggested biochemical mechanisms.

### Funding

The study was funded by an American Heart Association Scientist Development Grant (10SDG3990004) awarded to Thomas R. Cimato. The funders had no role in study design, data collection and analysis, decision to publish, or preparation of the manuscript.

### Grant Disclosures

The following grant information was disclosed by the authors:
American Heart Association Scientist Development: 10SDG3990004.

### Competing Interests

The authors declare there are no competing interests.

Peer**J**

## Author Contributions

- Thomas R. Cimato and Beth A. Palka conceived and designed the experiments, performed the experiments, analyzed the data, contributed reagents/materials/analysis tools, wrote the paper, prepared figures and/or tables, reviewed drafts of the paper.

## Human Ethics

The following information was supplied relating to ethical approvals (i.e., approving body and any reference numbers):

Informed consent to undergo the study protocol was obtained in writing from each study subject according to the principles expressed in the Declaration of Helsinki and approved by the University at Buffalo Institutional Review Board for Health Sciences Research (Approval Number: MED5980509B).

## Supplemental Information

Supplemental information for this article can be found online at http://dx.doi.org/10.7717/peerj.764#supplemental-information.

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
