# Peer review of "Effects of statins on TH1 modulating cytokines in human subjects"

_PeerJ, doi:10.7717/peerj.764_

## Round 0.1 · original submission · Major Revisions

There seems to be a consensus around the fact that the sample size might be too small and the statistical analysis in need of improvement. The authors did state that these was a preliminary study but I wonder if there is any way that the comments of the reviewers could be taken on board by changes in the texts and possibly a further statistical analysis?I have asked for major revision mainly to 'stimulate' visible changes in the text.

Reviewer 1 ·

Basic reporting

Several sentences in the introduction should be carefully checked. For example the authors report that IFN-gamma promotes atherosclerosis by inhibiting MMP activity. However, several data demonstrate that IFN-gamma increases protease secretion and activity. The lack of IL-27r and IL-27 p28 subunit expression has been shown to increase (and not attenuate) experimental atherosclerosis.

Experimental design

See below

Validity of the findings

The main aim of the manuscripts is to correlate the reduction of serum lipid levels induced by statins in healthy adult subjects with the level of circulating pro- and anti-inflammatory cytokines. As correctly stated by the authors, the study is limited by the small samples size of the cohort. Only 12 patients were analyzed with few cytokines not detectable in some subjects. Thus the study is underpowered to determine if the statin treatment affects the cytokine levels. The linear correlation analysis is performed analyzing a total of 34-44 samples, however still 11 or fewer subjects generate these samples. I am not sure this is the correct statistical approach. As such the data are too preliminary for publication.

Minor points:
Plasma levels of the analytes are not always measured in triplicate;
Data in Figure 6B refer to IL-12 and not IL-18BP as reported in the text.

Reviewer 2 ·

Basic reporting

This paper is well written.

Experimental design

See below.

Validity of the findings

These are rather preliminary results, but interesting as they stand.

Additional comments

Review of Cimato and Palka-
This is an interesting study which attempts to document the effect of cholesterol on selected Th1-associated cytokines through the use of statins. While the four proteins studied represent only a portion of the Th1 cytokines and associated proteins this is still a worthwhile study. The study is based on only 12 persons so that it is a very preliminary. This is also a straightforward paper with not much to say given that the results are preliminary. Please address the following points if you think it will strengthen the paper.

Point #1. Why choose the cytokines IL12p40 and IL18 in particular. Perhaps IFN-gamma is the most quintessential Th1 response cytokine.

Point #2. As the authors reason, statins affect the level of cholesterol which then affect the level of cytokines, is very reasonable. However there is also the possibility that the statins might directly affect the cytokines which then affect cholesterol. For example statins have been shown to up-regulate IL10. Then also IL10 has been shown to regulate cholesterol levels. So while I believe the authors are probably correct, technically there are other possibilities which should be mentioned. One way to deal with this may be to interchange “cholesterol” in the title with “statins” .

Point #3. Various statins have been shown to have variable effects on the levels of other Th1 cytokines produced by peripheral blood mononuclear cells, but there were considerably different effects seen for each statin tested (Bessler et al., 2005).

Reviewer 3 ·

Basic reporting

No Comments

Experimental design

This is an exploratory study on the potential effects of cholesterol on Th1 cytokines in human plasma of subjects without base inflammatory processes.

The study uses a limited number of subjects which gives a low statistical power to address the scientific problem.

However, the report is well conducted and authors found correlations between some cytokines and the plasma levels of total cholesterol and other bio markers of lipid metabolism. These correlations, although no strong, are statistically significant and perhaps biologically relevant as well.

From the Methods section, one can understand that all patients received each of the three statins. It is not clear why authors decided to give all statins to each patient in a particular sequence. While this allowed to test the performance of each statin regarding the study variables (cytokines) in the same individual, it may have induced authors to analyzed all sets of data for each cytokine by non-parametric ANOVA and Bonferroni post test. I am afraid this is not the appropriate analysis according to the study design.

It may be interesting if authors analyze their data with a two-group approach (Control vs each Statin) instead. This will allow to analyze in fact, what statin will have the best effect over the levels of each cytokines. Analysing IL-18 with Mann Withney test using the data provided by authors, I was able to find a statistical difference between control and Atorvastatin with a p value of 0.0186. This difference should be biologically relevant (Means 69.06 ± 9.626 vs 119.3 ± 18.65 respectively).

Hence, this reviewer recommends authors to perform a different analysis of cytokine levels among groups to compare in each treatment set, each cytokine versus untreated group by means of a Man Whitney test or other non-parametric test for testing differences between two groups.

Validity of the findings

No Comments

Additional comments

There are some typographical errors that need to be revised for clarity of the text. For example, Line 174 "statins treatment effects..."; line 61 "...no medical problems". A complete review of the text in this regard is recommended.

---

## Round 0.2 · accepted · Accept

I think that as long as the paper states clearly the limited number of samples and the relative conclusions that can be drawn upon it, then there is no reason why it should not be considered for publication.

Pilot data are always useful and pave the way for further exploration.